# Revisiting Deep Hybrid Models for Out-of-Distribution Detection

**Paul-Ruben Schlumbom**                                             *pschlumb@waikato.ac.nz*
*Department of Computer Science*
*University of Waikato*

**Eibe Frank**                                                       *eibe.frank@waikato.ac.nz*
*Department of Computer Science*
*University of Waikato*

**Reviewed on OpenReview:** *https://openreview.net/forum?id=yeITEuhv4Q*

## Abstract

Deep hybrid models (DHMs) for out-of-distribution (OOD) detection, jointly training a deep feature extractor with a classification head and a density estimation head based on a normalising flow, provide a conceptually appealing approach to visual OOD detection. The paper that introduced this approach reported 100% AuROC in experiments on two standard benchmarks, including one based on the CIFAR-10 data. As there are no implementations available, we set out to reproduce the approach by carefully filling in gaps in the description of the algorithm. Although we were unable to attain 100% OOD detection rates, and our results indicate that such performance is impossible on the CIFAR-10 benchmark, we achieved good OOD performance. We provide a detailed analysis of when the architecture fails and argue that it introduces an adversarial relationship between the classification component and the density estimator, rendering it highly sensitive to the balance of these two components and yielding a collapsed feature space without careful fine-tuning. Our implementation of DHMs is publicly available[1].

## 1 Introduction

Out-of-distribution (OOD) detection is crucial for reliable AI systems in safety-critical domains. Deep learning systems are known to exhibit "silent failures" yielding highly confident incorrect predictions (Guo et al., 2017), prompting the need for methods that recognise inputs outside the distribution underlying the training data. One such approach is to use density estimators to model this distribution. Normalising flows, which perform density estimation by establishing a diffeomorphism between the distribution of the training data and a well-defined base distribution, are flexible density estimators that seem very suitable for this. However, Nalisnick et al. (2018) observed that they often assign higher density to OOD observations than training data. Kirichenko et al. (2020) argued this is because they model the distribution in a low-level feature space when applied to pixels directly, indicating the need for approaches based on high-level semantic features. The deep hybrid model (DHM) for OOD detection (Cao & Zhang, 2022) follows this approach and reportedly achieves 100% OOD detection rates on common OOD detection benchmarks, a result that is cited as state-of-the-art (see, e.g., Humblot-Renaux et al. (2023)).

As software implementing DHMs for OOD detection is unavailable, we set out to replicate the approach. We obtain results comparable to other state-of-the-art methods but short of 100% AuROC. Indeed, similarly to Fort et al. (2021), we identify overlap between CIFAR-10 and CIFAR-100, rendering such a result questionable. We argue that the adversarial relationship between the classification component and the density estimator renders the DHM extremely sensitive to the balance of the loss signals from these two components.

---

[1] https://github.com/P-Schlumbom/deep-hybrid-models

This is a commonly recognised dynamic in the generative modelling literature; see, for example, (Xiao et al., 2019). We also argue that the proposed mechanism by which the DHM's feature extractor is supposed to preserve input volume in the feature space is problematic in practice, contributing to the DHM's brittleness.

Our contributions in this work are as follows:

1. We replicate the DHM and provide its code.

2. We find that performance is competitive with SOTA, but not better, and give reasons why 100% AUROC does not seem achievable.

3. We argue that considering the properties of the model, this is explained by an adversarial relationship between the classifier and the density estimator, and by a failure of the feature encoder to uphold the volume-preserving assumption posited by Cao & Zhang (2022).

## 2 Background and Related Work

Recent work in OOD detection in the context of classification, motivated by the arguments in Ahmed & Courville (2020), focuses on methods of detecting new classes of data exhibiting similar distributions of low-level features as the classes in the training data. OOD performance is typically measured with an OOD testing set containing different classes than the in-distribution (ID) training set. The CIFAR-10 and CIFAR-100 datasets (Krizhevsky et al., 2009), both sampled from the "80 million tiny images" dataset (Torralba et al., 2008), and exhibiting mutually exclusive classes, are a popular benchmark. Similarly, the SVHN dataset (Netzer et al., 2011) is often employed as benchmark OOD data for CIFAR-10. Normalising flows trained on CIFAR-10 often assign higher density to SVHN samples (Nalisnick et al., 2018).

Several approaches have been proposed to model the density in a higher-level feature space provided by a feature extractor. Lee et al. (2018) fit Gaussian distributions to the feature vectors of a trained classifier, while Feinman et al. (2017) applied a kernel density estimator. Liu et al. (2020) and van Amersfoort et al. (2020) advance this by applying spectral normalisation to the feature extractor's weights. Spectral normalisation, introduced in (Yoshida & Miyato, 2017), imposes an upper bound on a network's Lipschitz constant. In residual neural networks, setting this upper bound for residual connections in residual blocks will also set the lower bound for the Lipschitz constant of the block, a property known as bi-Lipschitz continuity (Liu et al., 2020). This is deemed particularly suitable for OOD detection as it yields *sensitivity* in the feature extractor, meaning that change in the input will yield a change in the corresponding feature vector, and *smoothness*, meaning the feature space reflects distances in the input space. van Amersfoort et al. (2020) and Postels et al. (2020) pointed out the tendency for "feature collapse" in extractors obtained using purely supervised training, where OOD samples share features with ID classes, rendering them indistinguishable in feature space. The latter argue that this can be mitigated by imposing sensitivity. They also introduce the concept of smoothness; Liu et al. (2020) refer to it as distance awareness.

Normalising flows provide a flexible approach to density estimation, making them attractive for OOD detection in classification models. This has been investigated using deep hybrid models, e.g., the architecture described by Zhang et al. (2020), which uses a feature extractor $\phi : x \mapsto h$ to generate features $h$ that are passed to a classifier head $C : h \mapsto y$ for classification and a flow-based model $F : h \mapsto z$ for density estimation, see Figure 1. Cao & Zhang (2022) adopted this approach, obtaining the method we refer to as DHM in this paper, by applying spectral normalisation to the feature extractor, similarly to van Amersfoort et al. (2020) and Liu et al. (2020). Use of spectral normalisation is motivated by the argument that a tight Lipschitz bound will approximately preserve volume in the transformation from $x$ to $h$. Sensitivity and smoothness are not considered. Notably, Cao & Zhang (2022) eventually loosen the Lipschitz bounds so as not to restrict the classifier's performance. Indeed, our results show that strong Lipschitz constraints usually disrupt classification performance in DHMs, even for simple synthetic datasets.

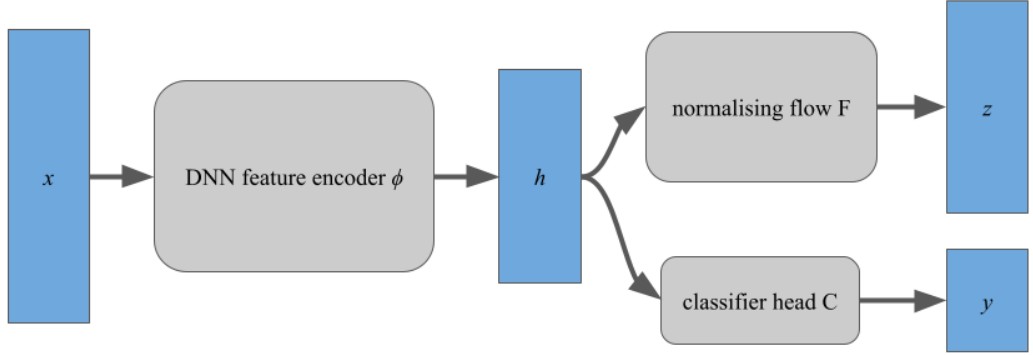

Figure 1: Diagram of the deep hybrid model architecture for out-of-distribution detection.

## 3  Implementing DHMs for OOD detection

The DHM's loss function combines the classifier (i.e., $C \circ \phi$) loss, $\mathcal{L}_{\mathrm{dnn}}$, and the normalising flow loss, $\mathcal{L}_{\mathrm{nf}}$. The classifier loss $\mathcal{L}_{\mathrm{dnn}}$ is the standard categorical cross-entropy loss. The normalising flow loss $\mathcal{L}_{\mathrm{nf}}$ is designed to maximise the likelihood of the training samples and is defined as:

$$\mathcal{L}_{\mathrm{nf}}(\theta) = -\frac{1}{N} \sum_{i=1}^{N} \log p_H^*(h_i), \tag{1}$$

where $\log p_H^*(h)$ is the log density assigned by the normalising flow to the sample $h = \phi(x)$. Note that Cao & Zhang (2022) argue that spectral normalisation in the feature encoder will ensure that $p(h) \approx p(x)$, and that estimating the density of $h$ therefore also provides the density for $x$. We question this claim in Section 4.4. For a normalising flow $f$ defined in the normalising direction (i.e., $f(h) = z$), and composed of $M$ diffeomorphic flow components, $p_H^*(h)$ is defined as

$$p_H^*(h) = p_Z(f(h)) \cdot \prod_{i=1}^{M} \left| \det \frac{\partial f_i(h_{i-1})}{\partial h_{i-1}} \right|, \tag{2}$$

where $h_i$ is the intermediate value $f_i \circ f_{i-1} \circ ... \circ f_1 \circ h$, with $h_0 = h$ and $h_M = z$, and $p_Z$ is some well-defined base density—typically a standard normal distribution. This is commonly stated as

$$p_H^*(h) = p_Z(f(h)) \cdot |\det(J_f)|. \tag{3}$$

Put together, the loss function for the DHM, $\mathcal{L}_{\mathrm{DHM}}$ is defined as

$$\mathcal{L}_{\mathrm{DHM}} = \mathcal{L}_{\mathrm{dnn}} + \lambda \mathcal{L}_{\mathrm{nf}}. \tag{4}$$

Since $\mathcal{L}_{\mathrm{nf}}$ can reach far greater magnitudes than $\mathcal{L}_{\mathrm{dnn}}$, and may overshadow it, it is scaled by the hyperparameter $\lambda$. The DHM proves to be highly sensitive to $\lambda$, as we explore further in Section 4.2.

Following the implementation described by Cao & Zhang (2022), we use a wide residual network (WRN) (Zagoruyko & Komodakis, 2016) as the feature extractor; specifically, the variant with 28 convolutional layers and a widening factor of 10, WRN-28-10.

Spectral normalisation is applied to the weight matrices of the encoder network. We follow the method described by Gouk et al. (2021), where the weights are left unnormalised if their spectral norm is below some coefficient $c$, a hyperparameter to be tuned. Cao & Zhang (2022) recommend using $c = 6.0$, although similarly to van Amersfoort et al. (2021), we find no degradation of performance even for $c = 3.0$ (see Figure 4).

We also apply $l_\infty$ normalisation to the feature vector. This step is not specified by Cao & Zhang (2022), but we found it to be vital for achieving good performance. We discuss why this is in Section 4.5.

For the normalising flow component, following Cao and Zhang, we use a residual normalising flow (Behrmann et al., 2019). We employ the implementation by Chen et al. (2019), applying it with 10 residual blocks (i.e., $M = 10$), each with a hidden dimension of 640. We leave out activation normalisation, as this yields better performance. The original work on deep hybrid models by Zhang et al. (2020) also describes preprocessing the features with a logit transform before passing them on to the normalising flow. Logit transforms were introduced by Dinh et al. (2016) to preprocess images for their RealNVP normalising flow and are defined as:

$$y = \text{logit}(\alpha + (1 - 2\alpha)\frac{x}{255}), \tag{5}$$

where the logit function is defined as $\text{logit}(p) = \log(p) - \log(1 - p)$, and $\alpha$ is some small positive value that prevents the $p$ from being exactly zero or exactly one. The purpose of the logit transformation is to map values from $(0, 1)$ to $(-\infty, \infty)$. In the case of modeling images, this is beneficial as it represents the bounded distribution of pixel values as an unbounded distribution, i.e., it prevents the normalising flow from producing impossible pixel values in the generative direction. However, since the normalising flow in the DHM is receiving the (theoretically unbounded) features of an encoder model rather than the image itself, the transformation is not appropriate, and we do not apply it. Indeed, keeping the logit transformation results in considerably worse performance.

We train the DHM components jointly, but with different algorithms for the weight updates. The feature extractor and classifier head are trained with SGD, using Nesterov momentum with a momentum of 0.9, an initial learning rate of 0.05, and a weight decay rate of 5e-4. The learning rate is scaled by 0.2 at 60, 120, and 160 epochs, and the model (unless specified otherwise) is trained for 200 epochs with a batch size of 256. The spectral normalisation coefficient $c$ is set to 3.0. The residual flow component is trained using Adam optimisation with a learning rate of 1e-4 and a weight decay rate of 16e-4. We chose different methods for the weight updates of the two components as we found them to respond better to these regimes when trained in isolation. Moreover, the DHM also seems to attain the best performance when the two different algorithms are used for the weight updates. When training for 200 epochs on an NVIDIA RTX 2080 Ti, this DHM configuration could be trained in about 17 and a half hours.

Additionally, uniform noise $u \sim U(0, 1)$ is added to the training images, a method introduced by Uria et al. (2013) for training normalising flows on images to prevent the model from assigning arbitrarily high density to discrete input pixel values. Although in DHMs, the normalising flow operates on extracted features, these are deterministic transformations of the original images, and we empirically observed that introducing noise is useful.

## 4 Experiments

We present results on the CIFAR and SVHN datasets for our implementation of DHMs. In Section 4.1, we cover our attempts at replicating the results from Cao & Zhang (2022). We investigate the effect of the hyperparameter $\lambda$ in Section 4.2. In Section 4.4, we study the volume-preserving characteristics of the DHM, and in Section 4.5, we explore the role of sensitivity in DHM performance.

### 4.1 Replication of DHM Results

Following the training regime described by Cao & Zhang (2022), we train the DHM model on CIFAR-10 training data and then evaluate the OOD detection performance on CIFAR-100 and SVHN test sets compared to the CIFAR-10 test set. This is done by computing the density assigned to each image by the DHM's normalising flow and then computing the area under the receiver operating curve (AuROC) and the area under the precision-recall curve (AuPR-in), where we treat ID samples as the positive class and OOD samples as the negative class. This is a standard evaluation procedure in the OOD detection literature, see Hendrycks & Gimpel (2016). Additionally, Humblot-Renaux et al. (2023) argue that AuROC and AuPR-in scores fail to capture the level of separation between the positive and negative sample populations and

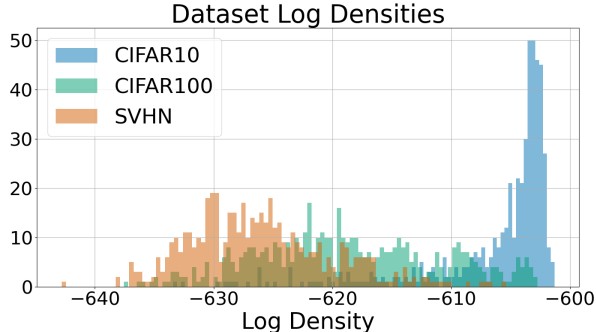

Figure 2: Histogram of estimated log densities for the DHM trained on CIFAR-10.

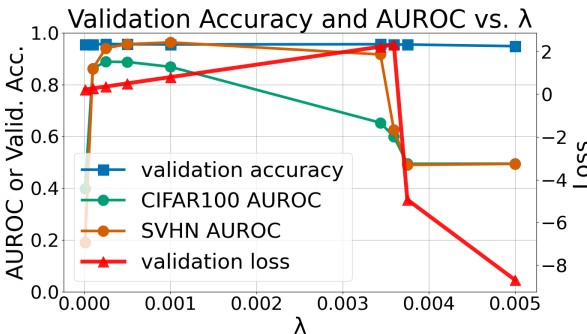

Figure 3: A plot of validation accuracy, validation loss, CIFAR-100 AuROC, and SVHN AuROC scores against $\lambda$, showing an immediate collapse of performance above a certain threshold.

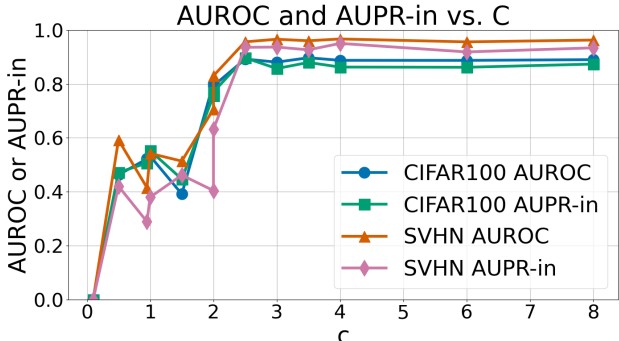

Figure 4: A plot of CIFAR-100 and SVHN AuROC and AuPR-in scores against the spectral normalisation coefficient $c$ for the DHM trained on CIFAR-10.

introduce the area under the threshold curve (AuTC) measure. The AuTC score becomes smaller as the separation of the populations becomes greater, and we report all three scores for our DHM implementation.

We report results for AuROC, AuPR-in, and AuTC in Table 1. For comparison, we also provide the results reported by Liu et al. (2020) for their SNGP architecture, which represents the state-of-the-art and is structurally similar to the DHM. We show the results for SNGP implemented with a WRN-28-10 feature extractor, as is used in our DHM implementation, to enable a direct comparison. We also include performance scores calculated with the classifier component's softmax scores as a simple baseline.

Our implementation of the DHM yields performance similar to the state-of-the-art in the field but is clearly not perfect. As is clear in Figure 2, the DHM is far from perfect in separating the ID sample densities from the OOD sample densities. Indeed, it is questionable whether a 100% OOD detection score is possible at all in the CIFAR-10/CIFAR-100 benchmark: as Fort et al. (2021) noted, there are several images in the CIFAR-10 dataset that in fact depict CIFAR-100 classes (see Figure 5), as well as ambiguous images such as vans that can be found in both the CIFAR-10 "car" class and the CIFAR-100 "bus" class. In performing a nearest neighbour search over the CIFAR-10 and CIFAR-100 images using the features of our DHM model, we also found the case of an identical image present in both datasets, and CIFAR-100 images that arguably depict CIFAR-10 categories (see Figure 6). Given the overlap between the CIFAR-10 and CIFAR-100 datasets, one would expect even an ideal model—with features that optimally exclude true OOD classes—to include OOD samples and exclude ID samples in the actual datasets used, and thus be unable to attain 100% OOD detection rates.

Table 1: Our DHM compared with results reported by Cao & Zhang (2022), SNGP (Liu et al., 2020), and the softmax OOD detection baseline (results averaged over 10 independent seeds). The CIFAR-10 dataset is used as the in-distribution dataset.

| MODEL | METRIC | CIFAR-100 | SVHN |
|---|---|---|---|
| DHM (ours) | AuROC ↑ | $0.897 \pm 0.008$ | $\mathbf{0.96 \pm 0.01}$ |
| | AuPR-in ↑ | $\mathbf{0.908 \pm 0.007}$ | $0.966 \pm 0.007$ |
| | AuTC ↓ | $\mathbf{0.344 \pm 0.006}$ | $\mathbf{0.27 \pm 0.01}$ |
| DHM (original) | AuROC ↑ | $1.000 \pm 0.00$ | $1.000 \pm 0.00$ |
| | AuPR-in ↑ | $1.000 \pm 0.00$ | $1.000 \pm 0.00$ |
| SNGP | AuROC ↑ | $\mathbf{0.960 \pm 0.01}$ | $0.902 \pm 0.01$ |
| | AuPR-in ↑ | $\mathbf{0.905 \pm 0.01}$ | $\mathbf{0.990 \pm 0.01}$ |
| Softmax Baseline | AuROC ↑ | $0.872 \pm 0.009$ | $0.95 \pm 0.01$ |
| | AuPR-in ↑ | $0.80 \pm 0.02$ | $0.93 \pm 0.02$ |
| | AuTC ↓ | $0.427 \pm 0.006$ | $0.36 \pm 0.02$ |

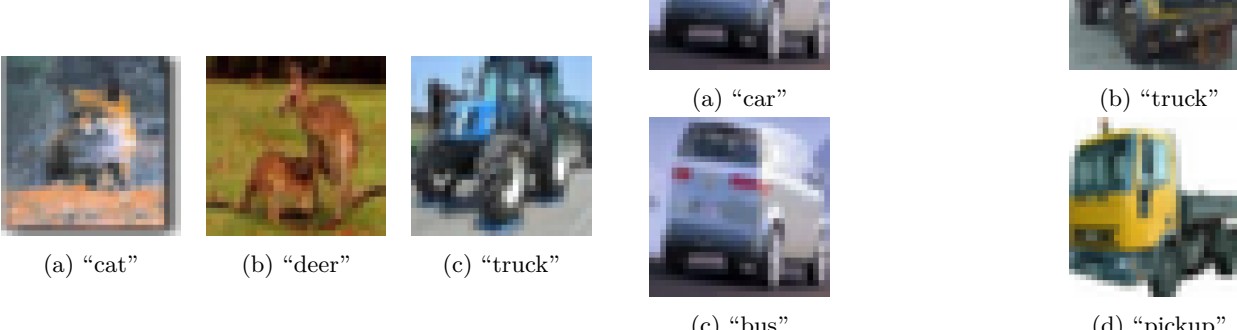

(a) "cat"    (b) "deer"    (c) "truck"

(a) "car"    (b) "truck"

(c) "bus"    (d) "pickup"

Figure 5: Images in the CIFAR-10 dataset that do not match their class.

Figure 6: Nearest neighbours between CIFAR-10 (top row) and CIFAR-100 (bottom row) samples.

## 4.2 Sensitivity to the Hyperparameter $\lambda$

The DHM proves to be highly sensitive to the $\lambda$ parameter in the hybrid loss function (see Equation 4). Since the log density calculated for the normalising flow is dependent on the determinant of the flow's Jacobian (see Equation 3), which can be arbitrarily large depending on the change in density between $x$ and $z$, the normalising flow loss can be considerably larger than the classifier loss. This can result in gradients obtained from that component of the loss overpowering those from the classification loss, causing the feature encoder to optimise for maximising $p_Z(f(x))$ by simply collapsing the feature space to maximise density of the observations in this space.

This can be seen clearly when measuring performance over $\lambda$ values (see Figure 3), where OOD detection performance immediately collapses above a certain threshold. Validation accuracy also begins to degrade above this threshold, albeit more slowly, so discrimination is still possible in the partially collapsed feature space. An inspection of the value of the final log determinant of the DHMs trained above and below the $\lambda$ threshold indicate that the feature space indeed collapses: moving from $\lambda = 3.594e - 3$ to $\lambda = 3.672e - 3$,

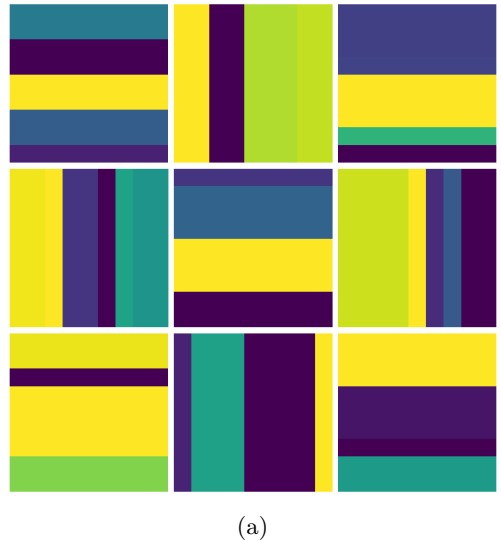
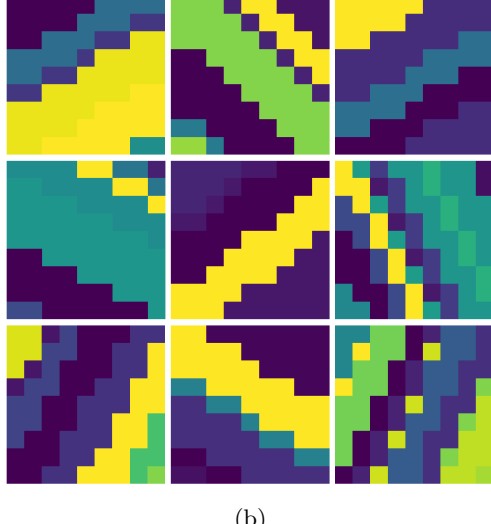

(a)                                                              (b)

Figure 7: Examples of the "stripes" dataset, with horizontal/vertical in-distribution samples and diagonal out-of-distribution samples.

the value of the normalising flow's log determinant jumps from 0.1209 to 1113. This shows that, after the threshold is exceeded, the change in volume is very large when moving between $z$ and $x$: the density for $z$ must be scaled by a large value to conserve total probability when transformed to the space in which $x$ resides because the feature values in $x$ are concentrated into a very narrow region.

### 4.3  Feature Collapse in Synthetic Data

We demonstrate the DHM's feature collapse directly with a synthetic dataset of 9×9 striped images, where vertical/horizontal stripes serve as the ID classes and diagonal stripes serve as the OOD samples. More specifically, we randomly generate images by creating stripes between 1-4 pixels wide and assigning to each stripe a value sampled uniformly from $[0, 1]$. For the OOD samples, the angle of the stripes for a given image is sampled from between 15° and 75° (i.e., $45° \pm 15°$), with the gradient flipped half the time to ensure diagonal stripes go in both directions. See Figure 7 for examples.

This synthetic dataset has several benefits for analysing the DHM. First, the distributions of low-level features (i.e., individual pixel values) are uncorrelated with the target classes: the target classes are only correlated with local patterns of pixel values (i.e., the gradient of pixel values along different axes), in other words, with higher-level features. This imitates the property we wish to capture in more complex natural datasets.

This also highlights the second benefit, that the "high-level" features are known, and can therefore be captured with a hand-crafted feature encoder. We use a simple 2-channel convolutional layer with horizontal and vertical Prewitt filters (Prewitt et al., 1970) and an absolute value activation function, followed by a maxpooling layer to generate a 2 dimensional output. Our DHM implementation uses the same architecture, leaving the convolutional parameters trainable. This allows us to compare the DHM's parameters after training to a known solution.

Finally, as the features are two-dimensional, they also enable direct visualisation for the benefit of inspection. The stripes dataset thus captures the relevant properties of natural datasets that OOD detectors are intended to capture, while doing so at an interpretable scale.

To create the DHM, we apply a softmax operation to the features to serve as the "classifier head" and attach a 50-layer residual normalising flow with a hidden dimension of 32 to serve as the density estimator. Note that while significantly smaller normalising flows could still reach the performance of the 50 layer model, they took longer to achieve the same performance in our experiments.

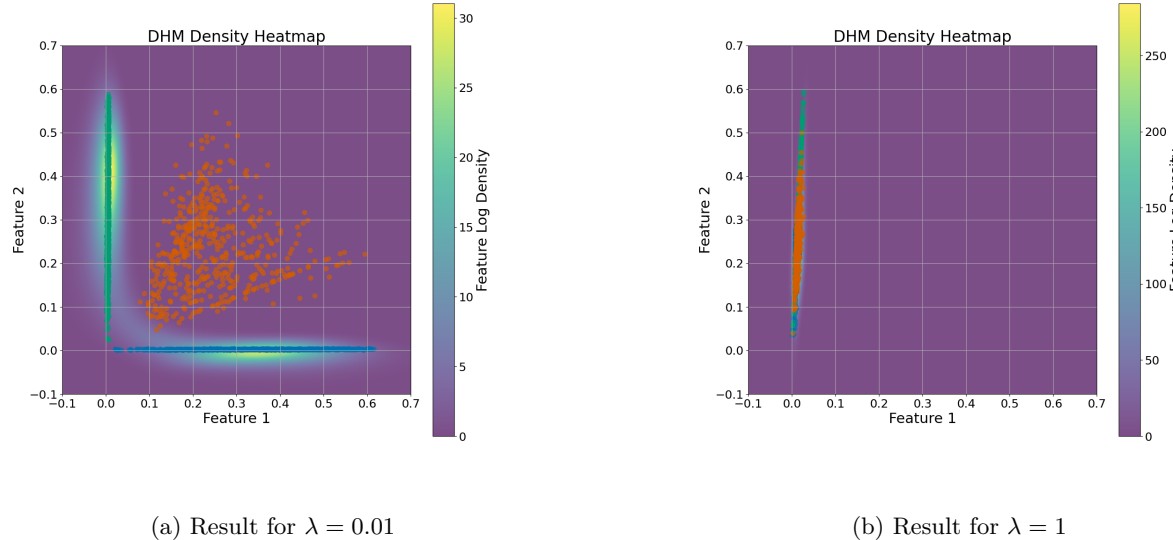

(a) Result for $\lambda = 0.01$                                          (b) Result for $\lambda = 1$

Figure 8: DHM density heatmaps for the stripes dataset, with datapoint embeddings overlaid, for low and high $\lambda$ values. Blue and green points are the horizontal and vertical ID samples, respectively, while orange points represent the diagonal OOD samples. This shows the collapse of features for high $\lambda$.

Upon conducting a search over $\lambda$ values, we see a repeat of the behaviour observed in Section 4.2, with OOD performance dropping to random guessing above a certain threshold. Visualising the features at two different $\lambda$ values (see Figure 8), we observe the features collapse into a single region that eliminates the distinction both between the ID and OOD samples and even the ID class samples.

## 4.4 Non-Volume Preserving Feature Spaces

For normalising flows applied to raw image data, feature collapse is not an issue as the datapoints are fixed; however, the DHM architecture allows $h$ to be adjusted to maximise $p_Z(z)$: the parameters of the feature extractor are free to be adjusted during training. To demonstrate this, we consider DHM architectures trained on small synthetic datasets with 2D features for ease of inspection. Since the purpose of the normalising flow is to establish a bijective mapping between the known density function $p_Z$ and the unknown density $p_H$, the density in the feature space preserves the volume of the base distribution and integrates to 1. We are interested in the volume of the density function integrated over the input space. In the case of the DHM with 2D data, we can pass the model a grid of points covering the input space and measure the density at each point to approximate the overall volume assigned by the DHM to the input space via the trapezoid.

We set up a simple DHM architecture where the feature encoder consists of a 6-layer Resnet with a hidden dimension of 128, and the normalising flow consists of a residual flow with 5 residual blocks and a hidden dimension of 128. By default, we use $c = 0.9$ as the value for the spectral normalisation coefficient as this empirically yields high accuracy. We train the model on the common "moons" dataset, with 2 classes, and a slightly more complex dataset of 5 concentric rings, where each ring is a separate class (see Figure 9).

We find the DHM assigns an approximate volume under the density of 0.4 across the input space on average and does not preserve volume. Indeed, Cao & Zhang (2022) explain that this will only be the case when the network's Lipschitz constant is close to 1, which can be enforced by setting $c \ll 1$ for a residual network (see Section 2). Unfortunately, in the degenerate case, this can result in a simple identity mapping. For this example problem, setting $c = 0.2$ yields an approximate volume of 0.995 for the input space but classification accuracy is reduced to 0.91 on the test set, from 1.0. We note that the same pattern arises when training

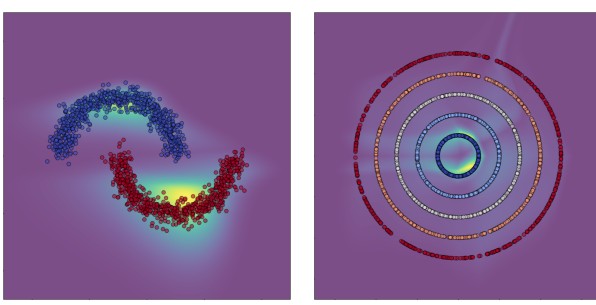

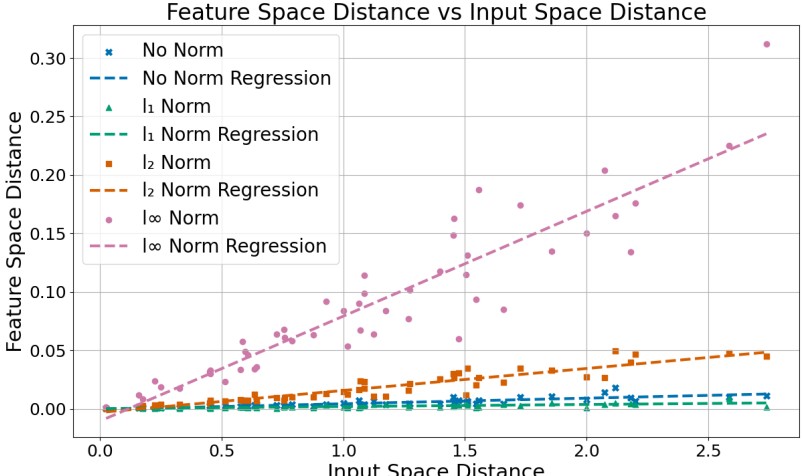

Figure 9: DHM densities assigned to moons (left) and rings (right) datasets, with the original datapoints overlaid.

Figure 10: Testing accuracy and approximate volume under the density on the synthetic rings dataset against the spectral normalisation $c$ coefficient. The horizontal dotted line denotes the expected volume of 1.

Figure 11: Deviations in feature space against deviations in input space, for the full DHM trained on CIFAR-10, showing that the $l_\infty$ normalised DHM is the most sensitive configuration.

the full DHM on the CIFAR-10 dataset, where OOD performance rapidly degrades for $c < 2.5$ (see Figure 4).

The phenomenon is more extreme on the rings dataset. Training the DHM with $c = 0.9$ yields a testing accuracy of 0.94, but the modelled density is concentrated on the central ring and has an approximate volume greater than 15. Constraining $c$ ensures more evenly distributed probability densities and an input space volume of $\sim 1$, but in exchange for a considerable drop in testing accuracy to 0.2 (see Figure 10).

## 4.5 Sensitivity and Normalisation in Feature Space

Our results indicate that rather than being dependent on the volume-preserving characteristics of the feature encoder, which was put forward as the motivation for spectral normalisation in Cao & Zhang (2022), the DHM's performance depends on feature spaces exhibiting smoothness and sensitivity, the properties considered by van Amersfoort et al. (2020) and Liu et al. (2020) (see Section 2).

It is instructive to consider the effect of different forms of normalisation of the feature vectors on sensitivity. By applying random deviations to a test input and measuring the resulting (Euclidean distance) deviations

Table 2: Summary of the normalisation schemes tested and the corresponding gradient and r-values. $l_2$ and $l_\infty$ normalisation shows the largest r-values (i.e., the strongest correlation between input and feature space deviations) and the greatest rate of change in feature space with respect to input space.

| NORM | GRADIENT | R-VALUE |
|------|----------|---------|
| None | 0.00478 | 0.847 |
| $l_1$ | 0.00176 | 0.693 |
| $l_2$ | 0.01879 | 0.930 |
| $l_\infty$ | 0.08966 | 0.921 |

in feature space, we can measure the sensitivity of a model directly. This reveals that the $l_\infty$ normalised configuration of the DHM (as described in Section 3) yields the greatest change in feature space for a given change in input space on average, and is therefore the most sensitive (see Figure 11). Furthermore, considering the correlation between feature space and input space distances (see Table 2, which shows the slopes of the lines in Figure 11 as well as the R-values between distances), we see that $l_2$ and $l_\infty$ norm yield the strongest correlation. This is evidence of 'smoothness'; that is, the feature space distances most strongly correlate with input space distances with these configurations.

This provides support for the use of $l_\infty$ normalisation in our experiments: it is likely to yield better OOD detection performance because it renders the method more sensitive to changes in the input and offers smoother feature spaces. This increases the separation of OOD samples from ID samples, making their identification a simpler problem, while ensuring these distances meaningfully correspond to changes in the input space.

## 5 Conclusion

In this paper, we revisited the deep hybrid model (DHM) architecture for OOD detection proposed by (Cao & Zhang, 2022). With our implementation of the algorithm, we reached performance comparable to the state-of-the-art but below the 100% OOD detection rates reported by (Cao & Zhang, 2022). This is consistent with observations made by Fort et al. (2021) and our own findings, namely that overlaps appear to make it infeasible to achieve perfect OOD detection rates on the datasets used in the experiments.

Our results show that the adversarial relationship between the classification head and the normalising flow under the joint training framework necessitates careful fine-tuning of the hyperparameter $\lambda$ to balance the contributions of the two components in the hybrid loss. We also found that the DHM's spectral normalization does not reliably preserve the volume of the input space, which calls into question the motivation for applying it put forward in (Cao & Zhang, 2022). We argue that the DHM's behaviour is better understood using the criteria considered by van Amersfoort et al. (2020) and Liu et al. (2020); that is, in terms of the feature encoder's *sensitivity* and *smoothness*.

While the DHM offers some promising results, it also proves to be fairly brittle. The fundamental issue it must contend with is the adversarial relationship between the classifier and the normalising flow, which encourages the collapse of features without careful fine-tuning. This suggests future work on alternative structural approaches that avoid this unstable behaviour would be beneficial. We note that there is some overlap with the unsupervised learning literature, where there is considerable interest in developing non-contrastive feature learning methods that do not degenerate to a trivial collapsed feature representation. For example, there has been some recent success with self-distillation methods (Grill et al., 2020; Caron et al., 2021). Future work may look at adopting some of these concepts to develop a more robust density-based OOD detection system.

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
