# OpenReview forum: "Revisiting Deep Hybrid Models for Out-of-Distribution Detection"
_TMLR — Accepted by TMLR_

### Review · Reviewer_gD8q · 2025-02-04

**Summary Of Contributions:**

Deep hybrid models (DHMs) are an approach to out-of-distribution (OOD) detection which simultaneously trains a feature encoder, along with a normalizing flow and a classification head on top of these features; the latter two can then be used for OOD detection. The original DHM paper reported perfect performance (AUC=1) at many common OOD tasks but provided no code. This paper deconstructs the original DHM paper: First, the authors attempt to reproduce results, and are unable to do so. Although the authors show that DHMs obtain highly competitive performance at OOD detection, they did not manage to obtain perfect AUC scores. The authors then create a toy dataset/OOD detection task to probe into the inner workings of DHMs, finding that the reason behind their good performance is likely not what was suggested in the original DHM paper. Along the way, the authors find that the performance of DHMs can be very sensitive to hyperparameter choices.

Overall I think this is a good submission which fits the acceptance criteria for TMLR: the paper will be of interest to the OOD detection community, and the claims made in the paper are backed by evidence since the experiments are well executed and accompanying code is provided.

**Audience:**

Yes

**Broader Impact Concerns:**

I have no broader impact concerns.

**Claims And Evidence:**

Yes

**Requested Changes:**

See weaknesses above.

**Strengths And Weaknesses:**

## Strengths

1. The paper is well written.
2. The paper is quite convincing is showing that the results in the original DHM paper are likely not reproducible.
3. The results in the paper are of interest to people working in OOD detection.

## Weaknesses

1. Reading the paper seems to convey the "adversarial relationship" between the feature extractor and the flow as a novel finding. It is well known in the generative modelling literature that simultaneously training autoencoders along with generative models in the corresponding latent space will typically exhibit these "adversarial relationship", see e.g. the discussion at the end of section 5.3 in [1] or section 3.3 in [2]. This is essentially the same phenomenon, and so exploring it in depth in the toy example seems slightly superfluous and should be better motivated, and wording which makes it sound like a novel finding should be toned down.
2. Although the paper is mostly well written, I do think there a few improvements that could be made. I think the biggest thing is that the description of the flow in section 3 could be cleaner: please be more specific about what $f$ is in terms of the notation that you have already introduced. I think it should be $f=F \circ \phi$, which would then imply that $h$ has the same dimension as $x$ but as someone who was not familiar with DHMs, it was unclear if $h$ was lower-dimensional and ignored in the computation of the log determinant; it was also unclear to me how the log determinant is computed, since it has to be computed not only for the flow but also for the encoder. Lastly, small details:
- Figure 1: please add function names to blocks ($\phi$ to the encoder, $F$ to the flow, and $C$ to the classifier). Also, please make the box for $h$ the same size as the box for $x$ to convey they are of the same dimensionality.
- Please use $\partial$ instead of $\delta$ in eq 2, it's much more standard notation.
- Please be more precise when appropriate in figure and table captions as sometimes the results being shown can only be understood when reading the main text. For example, Table 1 does not mention the training set being used, figure 10 does not explain if it is showing results on toy or image data, table 2 does not even say what metric is being shown, etc.
- The captions in figure 1 and table 2 are missing periods at the end.
- The equations in the legend of figure 10 are highly uninformative.
- Please use \citet and \citep appropriately in LaTeX, I found a few incorrect uses.
- "< <" should be replaced by "\ll" in LaTeX


[1] Deep Generative Models through the Lens of the Manifold Hypothesis: A Survey and New Connections, Loaiza-Ganem et al., TMLR 2024

[2] Generative Latent Flow, Xiao et al., arxiv 2019

---

> ### Author Response · Authors · 2025-02-25
> **Regarding adversarial dynamics, notation, and density estimation assumptions**
>
> Thank you for your feedback.
> Regarding your points:
> 1. We argue that the adversarial relationship is one of the causes of the DHM's sensitivity to hyperparameter tuning, but this is not intended as a generally novel result: rather, it provides an empirically backed explanation for why the DHM architecture is so sensitive to hyperparameter choice.
> We have rewritten the relevant section in the introduction to make this clearer: “We argue that the adversarial relationship between the classification component and the density estimator renders the DHM extremely sensitive to the balance of the loss signals from these two components. This is a commonly recognised dynamic in the generative modelling literature; see, for example, (Xiao et al., 2019).” to emphasise that this dynamic is an explanation for the observed behaviour, not a novel discovery.
>
>
> 2. In Figure 1, $h$ is meant to be the same size as $z$ rather than $x$, so we have kept it the same. This addresses the issue raised regarding the normalising flow notation; we agree that using $x$ to denote the input-space variable, which is the typical notation for normalising flows, is confusing in this context, where we actually use the intermediate $h$ features as input for the normalising flow. We have updated the notation to reflect this.
>
> As to how the log determinant for the encoder is computed: the original paper argues that the introduction of spectral normalisation will ensure that the encoder approximately preserves volume such that $p(x) \approx p(h)$, so that computing $p(h)$ also yields $p(x)$. This assumption of volume preservation is something we call into question; we agree that this raises the question of how to properly estimate $p(x)$.
>
> We have addressed the formatting issues you raised and added more details to several of our captions.

---

> > ### Comment · Reviewer_gD8q · 2025-02-26
> > **Thanks for the clarification**
> >
> > Thank you very much for the updates and added clarification, I had indeed originally misunderstood the normalizing flow computations and I think the latest updates are much clearer. I have also read through the other reviewers' reviews and maintain my original assessment of the paper. My last two nitpicks are that after equation 1 you wrote $h = \phi \circ x$ yet $x$ is not a function and so this should be replaced by $h = \phi(x)$, and another incorrect usage of \citep in the last bullet point in the introduction.

---

### Review · Reviewer_TFSm · 2025-02-12

**Summary Of Contributions:**

The paper reproduces and analyzes an existing out-of-distribution detection algorithm, namely the Deep Hybrid Model (DHM) of (Cao & Zhang, 2022), of which there exist no public implementations. In their analysis, the authors notice an inverse relationship between classification and density estimation. The paper also presents other details for optimizing performance, e.g. tuning the loss coefficient hyperparameter, normalization of the feature vectors, etc.

Questions:

Could the authors please enumerate the contributions of this work, e.g. in the intro, abstract, or conclusion, and emphasize what they see to be the primary contributions?

**Audience:**

Yes

**Claims And Evidence:**

No

**Requested Changes:**

1. Could the authors please enumerate the contributions of this work, e.g. in the intro, abstract, or conclusion, and emphasize what they see to be the primary contributions?
2. For claims made (e.g., identifying an adversarial relationship between classification and density estimation), could the authors ensure sufficiently supporting evidence is provided, e.g., experiments over multiple datasets and tasks?

**Strengths And Weaknesses:**

Strengths:
1. The paper is easy-to-read.
2. The background provides good motivation for the analysis.
3. The careful reporting of experimental details is much appreciated.

Weaknesses:
1. The claims could be better substantiated. For instance, a main contribution claimed by the paper is identifying an adversarial relationship between classification and density estimation, but specifically in individual use cases mentioned in the text rather than more systematically. Moreover, given Cao & Zhang 2022 present plots which do not show an obvious inverse correlation (e.g., Figs 4 and 5), a more substantiated claim would ideally explain how their previous results fit into the picture presented in this work.
2. It's hard to pinpoint the precise contributions of the analysis beyond observations on a specific dataset / setting. While there seem to be various details in this paper not mentioned in the original, their presentation is rather diffuse and specific to a particular example, making it hard to appreciate whether they are unique to the specific setting run or more systematic. For instance, while many of the figures are interesting, they seem more illustrative (e.g., $\ell_\infty$ normalization performs better than others on a dataset) than a comprehensive experimental contribution.

---

> ### Author Response · Authors · 2025-02-25
> **Clarifying contributions and adversarial dynamics**
>
> Thank you for your response. Regarding your requested changes:
> 1. We have added points summarising our contributions in the introduction.
> 2. Regarding the adversarial relationship between classification and density estimation, reviewer gD8q has pointed out this is a commonly recognised dynamic in generative modelling. We have shown that controlling the magnitude of the influence of the CE and flow loss on the final loss determines OOD detection performance. Furthermore, we would argue that the original Cao & Zhang 2022 figures 4 & 5 do show the behaviour we have also identified: AUROC scores are shown to drop significantly for very small $\lambda$ (as our Figure 3 shows), and also begin to degrade with increasing $\lambda$; our figure additionally shows AUROC scores degrading before reaching a sharp dropoff point for sufficiently large $\lambda$. We believe that their plots, over a greater range of values for $\lambda$, would show a similar sharp drop in AUROC scores above a certain threshold.

---

### Review · Reviewer_xiyb · 2025-02-13

**Summary Of Contributions:**

- The authors attempt to replicate the DHM results found in Cao & Zhang 2022. This is a relevant and interesting study because that paper claimed perfect scores on a standard OoD performance benchmark.

- The authors quickly note three pathologies with the Cao & Zhang work:
    - L_inf normalization, which was not used in the initial study, is critical for good performance
    - Logit transforms, which were used in the original study, result in significant performance decreases
    - CIFAR10/100 have duplications and errors that preclude 100% separation of these datasets in an OoD task

- Additionally:
    - classification performance is inversely related to input-space volume preservation under this method, illustrating that it has a fundamental limitation and further undermining the original papers claims (and also raising questions about volume preserving motivations for spectral normalization)
    - The authors argue that discriminating samples via feature space relies on properties of sensitivity and smoothness over that feature space (previously proposed by van Amersfoort et al. (2020) and Lie et al. (2020)

**Audience:**

Yes

**Claims And Evidence:**

Yes

**Requested Changes:**

Not critical to acceptance but please:

1) It would be nice to have a plot of AUROC / AUPR changes with respect to C, the spectral norm coefficient.
2) Normalizing flows can be quite slow to train. Please comment on the training resources used and how long it took, as that would further illuminate the utility (or lack thereof) of the proposed method
3) a couple of typos: "cleary" should be "clearly" in last paragraph of 4.1, "content" should be "contend" in last paragraph of conclusion

I would appreciate your comments on the below, as I think these ideas would strengthen the paper, but they are not necessary for acceptance:

- Neural collapse metrics would be useful. Papyan et al. and others have studied the relationship between CE loss and the resultant simplex ETF geometry of feature space. There is a “compressive” effect here, and it is also known that contrastive loss approaches simplex ETF geometry, but mitigates intraclass variability (Fang and He et al, 2021), and this is probably why contrastive learning can produce discriminative features well suited for OoD and classification purposes.
- It would be interesting to study what is happening to the simplex ETF geometry that CE loss will pursue while also trying to maximize P(z). Is flow loss “breaking” the simplex ETF structure significantly?
- Flows are universal density approximators in theory, but employ a litany of shortcuts and approximations to manage the O(d^3) complexity due to the Jacobian. In my experience, a single flow trying to measure density over a simplex ETF with 10 modes would be tricky for most if not all implementations, let alone what ends up happening with high-dimensional issues (Nalisnick et al. 2018), but I’m open to push back on this as I haven’t tried to solve this extensively. I would be interested in knowing how OoD performance and the adversarial nature of flow/CE loss is tied to the residual flow implementation. This is especially interesting, I think, because L2 normalization of feature space will, of course, constrain to a hypersphere, and max-norm to a hypercube, and certain flow implementations exist to deal with the former, if not the latter as well.
- In summary:
    - I’m curious whether the adversarial relationship is less due to a trivial solution for flows (although I’m sure this is still in there) and more to do with a regularizing effect due to the flow implementation, and how is this related to the simplex ETF optimum of CE loss?
    - I too like the “sensitivity and smoothness” way of conceptualizing feature space in this domain. I think Neural Collapse may shed some light on the shape of the manifold we end up dealing with under CE loss and maybe even suggest ways to characterize things.

**Strengths And Weaknesses:**

Strengths:

- The paper addresses a method that made strong claims, but for which there is no known implementation. This is valuable info for the OoD community.
- The claims are supported by experiments.

Weaknesses:

- There are some theoretical angles and experiments that could strengthen the paper, I comment on these below.

---

> ### Author Response · Authors · 2025-02-25
> **Corrections, ETF score and training normalising flows on manifolds**
>
> Thank you for your feedback. As suggested, we have made the following changes:
> We have added a figure showing AUROC/AUPR vs C (Figure 4), which shows performance falling off for c < 2.5.
> We have also added a note on training resources and training time in Section 3.
> We have corrected several typos, thank you for pointing them out.
>
> The neural collapse metrics - in particular, the simplex ETF geometry measure - is indeed very interesting. While we did not measure simplex ETF geometry during our primary experiments, some experiments on smaller models over shorter training runs in response to this suggestion indicate that the DHM features actually converge slightly faster to a simplex ETF than when training the encoder with pure CE. Perhaps more importantly, we also notice that, when we train a DHM model with high $\lambda$ to induce feature collapse, the simplex ETF measure fails to converge and remains high, indicating that the class mean vectors are failing to diverge. We interpret this as further evidence of feature collapse supporting our results in the paper.
>
> Regarding concerns about the normalising flow implementation, it is indeed the case that variants have been developed for operating on manifolds, including hyperspheres. We focused on modelling in Euclidean space because that is what the original paper explicitly specifies. Prompted by the comment, we did try a very simple normalising flow modelling a hypersphere manifold instead, and saw neither major advantages nor disadvantages - but did not conduct a thorough hyperparameter search. It is possible that this avenue could yield superior results.

---

### Decision · Action_Editor_dHa8 · 2025-03-17

**Recommendation:** Accept with minor revision

**Comment:**

This paper revisits the Deep Hybrid Models (DHMs) approach for out-of-distribution (OOD) detection, originally introduced by Cao & Zhang (2022), by carefully reproducing and analyzing its claims.  Through careful reproduction and additional analysis, this work identifies critical implementation details that significantly affect model performance, providing valuable insights and public implementations beneficial to the OOD detection community. The reviewers broadly agree on the value of the careful experiments and analysis, despite minor concerns about the breadth of datasets used and the clarity.

For the final version, please make sure to update the link to the public code.

**Audience:**

The findings presented in this paper will undoubtedly be of interest to a substantial portion of TMLR's audience, particularly researchers and practitioners working in the area of out-of-distribution detection and generative modeling. Given the growing interest in reliable OOD detection methods, the audience will benefit significantly from the careful analysis and practical insights offered by this paper.

**Claims And Evidence:**

The claims made in the submission are well supported by accurate, convincing, and clear evidence. The authors thoroughly examine the Deep Hybrid Model (DHM) originally proposed by Cao & Zhang (2022), addressing previously unreported sensitivities to hyperparameters and feature normalization.